# Does the risk of major customer need to be balanced? The role of customer concentration in corporate governance

Di Gao[1], Jiangming Ma[2]*, Yiru Wang[3]

**1** School of International Business, Southwestern University of Finance and Economics, Chengdu, Sichuan, China, **2** School of Economics, Xihua University, Chengdu, Sichuan, China, **3** School of Business, Cangzhou Normal University, Cangzhou, Hebei, China

* mjm121121@163.com

**Data Availability Statement:** All relevant data are within the manuscript and its Supporting Information files.

**Funding:** This work was supported by Fundamental Research Funds for the Central

## Abstract

In the operation and management of the company, major customers may affect a supplier firm's level of governance. The goal of our study is investigating whether a major customer acts as an important role in corporate governance in emerging markets and exposing the mechanism that how major customers affect corporate decision-making. There is a growing body of literature involving studies about the effect of customer concentration on firm performance of western countries. Few studies have recognized to what degree does customer concentration satisfy the sustainable development of supplier firm. Using a sample of Chinese listed firms, we found a nonlinear relationship between customer concentration and risk-taking, corporate policies and firm performance. Evidence shows that the effect of customer concentration in China resembles an inverted U-shaped curve and major customers are crucial in financial and investment policies. Our results help to provide a broader perspective on the role of major customers, giving a deep explanation about the role of customer concentration in corporate governance.

## 1. Introduction

Supply chain stability is not only advantageous to the integration of supply chain resources and information sharing between upstream and downstream operations, but it also acts as an external governance mechanism to improve supplier performance [1]. Major customers may affect a supplier firm's level of governance through two major mechanisms: the certification hypothesis and the concentrated credit risk hypothesis [2]. On the one hand, customer concentration has been proven to provide incremental information for investors and other stakeholders and affect stock price [3], cost of equity capital [4], bank credit [5], auditor decision-making [6], and profitability of an enterprise [1]. Due to the advantage of supply chain resources integration and information sharing operations, they also have strong incentive to monitor their suppliers to reduce risk of cooperation [7]. On the other hand, the higher the customer concentration, the riskier the reliance on major customers becomes. Hence, suppliers are more likely confronted with customer bargaining power over purchase prices [4, 8] and

Universities No. JBK 2101036, Modernization of Urban and Rural Governance Research Center of Chengdu Key Research Base of Philosophy and Social Sciences No. CXZL202102, Key Research Base of Philosophy and Social Sciences for Colleges and Universities in Sichuan Province No. KJJR2019-004 and Talent Introduction Project of Xihua University No. w202247.

**Competing interests:** The authors have declared that no competing interests exist.

operation risk and financial distress cost [9, 10]. This scenario could result in potential liquidity risk and switch costs for suppliers. Thus far, it is very difficult to maintain a customer relationship that is both concentrated and satisfactory.

Evidence shows that corporate risk-taking is valuable for obtaining additional profits and long-term development for firms, it could influence corporate decision-making behavior and corporate governance [11]. Besides, risk-taking must be maintained at a moderate level. If its level is remarkably high, firms risk insolvency or even bankruptcy. Conversely, if its level is remarkably low, it will lead to the lack of innovative resources investment. Therefore, improving the level of risk-taking and avoiding excessive risk-taking have major concern in the field of corporate governance. Prior research shows that corporate risk-taking behaviors are not only influenced by managers characteristics, such as CEO religion [12], optimism and risk aversion [13], political preferences [14], gender [15], and social capital [16], but is also affected by the ability to acquire resources and the external stakeholders [17, 18], literature mainly highlights the impact of institutional investors, creditors, and employees on corporate risk-taking [19–21]. In fact, stakeholders include both the investing individuals (shareholders) and the non-investing individuals (employees, suppliers, customers, community, etc.). Customer–supplier relationship, as a social network, has the effect of resource allocation, which can help enterprises more easily obtain additional resources at a low cost [2, 22].

In this study, we investigate the effect of customer concentration on corporate risk-taking, financial and investment policies, and firm performance. In particular, we ask the following question: Does the risk of major customers need to be balanced? How does the concentrated customer affect corporate risk-taking? The goal of this study is to examine the impact of customer concentration on risk-taking and corporate governance in emerging markets. We have selected China, an emerging economy, and Chinese-listed firms which generally rely on their major customers and suppliers, providing adequate sample variation to test the arguments. Nearly half of the firms do more than 30% of their business with the top five customers or suppliers, and about 20% do more than 50%. The concentration of customers and suppliers indicates that the firm's investment activities are highly proprietary, resulting in challenges to the risk of different firm decisions. The relationship between Chinese firms and stakeholders not only significantly differs from that of Western developed countries, but also the institutional environment that Chinese enterprises face is fundamentally different from that of Western countries [23].

Our baseline results show that supplier firms with low customer concentration are more likely to take higher risks for supply chain stability, while those with high customer concentration are more likely to take less risks. First, we observe an inverse U-shaped curve exists between customer concentration and corporate risk-taking. Second, we examine the effects of customer concentration on corporate policies. In terms of financial policies, customer concentration and financial leverage demonstrates an inverted U-curve relationship, whereas a U-shaped relationship is observed between customer concentration and asset liquidity. Simultaneously, a prominent inverse-U shape relationship is observed between customer concentration and R&D investment and capital expenditures. Finally, we provide sufficient support for the inverted U-shaped relationship between customer concentration and firm performance. The main results of our additional tests show that customer concentration is an important factor contributing to corporate risk-taking and corporate governance. Our results are robust to alternative proxies and endogeneity test.

Our study makes several contributions to the related areas of the literature. First, this paper provides new insight into the determinants of risk-taking in China. Unlike previous studies, which mainly focus on a managers' characteristics [12–16], external governance [17, 19–21], this study examines the impact of a significant external stakeholder: the role of major

customers in corporate governance. Our analysis highlights the importance of major customers in a firm's risk-taking behaviors in emerging markets. Second, this study explores the mechanism of how major customers affect financial and investment policies, giving an explanation about the ambiguous results of previous literature investigating only a linear relationship between customer concentration on corporate decision making [2–7, 10, 24]. In particular, the relationship between firms and customers in China is different from that of Western countries owing to their different institutional environments. Through empirical analysis, the effect of customer concentration in China perfectly resembles an inverted "U", which is consistent with the hypotheses about major customer function with different levels of customer concentration (certification hypothesis and concentrated credit risk hypothesis).

The rest of the paper is organized as follows. Section 2 shows the literature background and hypotheses development. Section 3 describes data and methodology. Section 4 examines the relation between customer concentration and risk-taking behavior, corporate decision making and firm performance, providing mechanism analysis. Section 5 addresses endogeneity problems and robustness checks. Section 6 offers conclusions.

## 2. Literature review and hypotheses development

### 2.1. Customer concentration and corporate risk-taking

Corporate risk-taking behavior is valuable for obtaining additional profits and long-term development for firms [11]. According to the agency theory, the management has the motivation of risk aversion [25]. To avoid loss of personal wealth, dismissal risk, and professional reputation caused by investment failure, the management may give up on positive net present value investment projects with high risk and adopt conservative investment strategy, which will undoubtedly damage the long-term value of the firm [11]. Research shows that corporate behaviors are influenced by managers' backgrounds and personal characteristics, which matter for corporate risk-taking, such as CEO religion [12], optimism and risk aversion [13], political preferences [14], gender [15], and social capital [16].

The risk-taking behavior does not only depend on the willingness of the management but is also affected by the ability to acquire resources and the external environment [17]. Boubakri et al. [18] study the impact of shareholders' identity on corporate risk-taking behavior and show the important role of state and foreign owners. From the perspective of corporate external governance, literature mainly focuses on the impact of institutional investors, creditors, and employees on corporate risk-taking [19–21]. For instance, Acharya et al. [19] debate whether creditor rights in bankruptcy affect corporate investment choice through corporate risk-taking. They find that an increase in creditor rights reduces risk-taking behavior. John et al. [11] demonstrates the importance of corporate governance choice in risk-taking, while Boubakri et al. [18] investigate how shareholders' identity, such as state and foreign owners, affects corporate risk-taking behavior. Guan and Tang [21] examine whether labor characteristics matter in corporate decisions, predicting corporate employees' risk attitude into corporate policies.

Recent studies on the effect of customer concentration or customer–supplier relationships have emerged, which have been shown to be related to earnings management [26], conservatism [27], cost of capital [4], mergers and acquisitions [28], innovation outcomes [29]. A large and growing body of literature has contributed to our understanding whether and how customer–supplier relationship affects the decision making of a firm. Hui et al. [27], Johnson et al. [30], and Cen et al [31] argue that the value of suppliers could be affected by major customers. There are two arguments that describe the role of major customers in the relationship of customer concentration and firm value, including the certification hypothesis and concentrated credit risk hypothesis.

Through a monitoring and certifying channel, suppliers benefit from reducing information asymmetry and effective access to the external resources required. Several researchers find that stable customer relationship help suppliers achieve improved inventory management, greater operational efficiency, and new product success, allowing firms to allocate more resources for value creation [3, 24]. For example, Wang et al. [9] show that major customers play a decisive role in dividend policy. Ahern and Harford [28] consider that a network of actual economic transactions helps explain the formation and propagation of merger waves as mergers propagate in waves across the network through customer–supplier relationships.

Conversely, others argue that a customer bargaining power could raise hold-up problem and result in higher concentrated credit risk and switching costs [4, 10, 32, 33]. If the customer concentration is remarkably high, suppliers with major customer-supplier relationships often must undertake relationship-specific investments, possibly resulting in higher concentrated credit risk and switch costs [9, 34] and lower profitability [1]. The relationship-specific investment requested by the major customers make the supplier more conservative in financial policy, such as paying fewer dividends [9]. Dhaliwal et al [4] state that customer concentration is related to higher systematic and idiosyncratic risk. Moreover, evidence suggests that strong customer bargaining power leads to hold-up problems and forces suppliers to invest less in R&D and innovation. In order to avoid risks, the level of risk-taking is more likely to be reduced. So, we put forward the competing hypothesis:

**Hypothesis 1A (H1A).** An increase in customer concentration could have a significant positive impact on firm's risk-taking.

**Hypothesis 1B (H1B).** An increase in customer concentration could have a significant negative impact on firm's risk-taking.

## 2.2. Customer concentration and corporate investment policies

Investment risk is most commonly used to measure the level of risk-taking of enterprises, and long-term strategic investment is more dangerous than short-term tactical action [35]. Different from the general operation and management activities, the innovation activities of enterprises have the characteristics of long cycle, high uncertainty of output and high adjustment cost [36]. Besides, Vogt [37] argued that capital expenditures are strongly and positively related to liquidity. Aggressive capital expenditures indicate a high level of liquidity risk of the firm. Although some researchers have found that R&D and capital expenditure have different implication for short-term profits. Both R&D and capital expenditure have positive impacts on the long-run success of a company [38–40]. Therefore, the higher R&D and capital expenditure related to investment policies, the higher the level of enterprise risk-taking. Building upon the relationship between R&D/capital expenditure and risk-taking, as discussed earlier, we concentrate on the impacts of customer concentration on R&D and capital expenditure. In the light of the certification hypothesis and the concentrated credit risk hypothesis, H2A, along with its opposing hypothesis, H2B is formed:

**Hypothesis 2A (H2A).** An increase in customer concentration could have a significant positive impact on firm's R&D (capital expenditure).

**Hypothesis 2B (H2B).** An increase in customer concentration could have a significant negative impact on firm's R&D (capital expenditure).

## 2.3. Customer concentration and corporate financial policies

The investment policies and financial policies are key elements of risk-taking behaviors and corporate governance. Begley et al. [41] find that CEOs could increase riskiness of financial policies

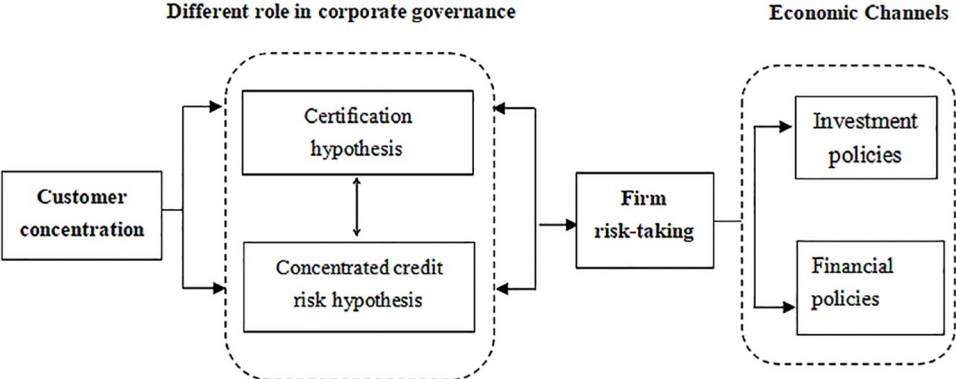

**Fig 1. Logical diagram of customer concentration effect on firm risk-taking.**

by holding less liquid assets or by increasing debt burden of the firm. Moreover, Ferris et al. [16] also show that riskiness of financial policies could be reflected in financial leverage and asset liquidity (working capital). External debt financing is more risky than internal financing, and the lower the debt financing is, the stronger the financial prudence is [42]. Therefore, the higher the debt, the higher the level of risk-taking. In other words, the higher R&D and lower working capital related to financial policies, the higher the level of enterprise risk-taking Thus, we expect a positive relation between customer concentration and financial leverage and negative association between customer concentration and asset liquidity (working capital). Based on these arguments and references, we develop a hypothesis, H3A, along with its competing hypothesis, H3B:

**Hypothesis 3A (H3A).** An increase in customer concentration could have a significant positive impact on firm's financial leverage and a negative impact on working capital.

**Hypothesis 3B (H3B).** An increase in customer concentration could have a significant negative impact on firm's financial leverage and a positive impact on working capital.

In order to explain the basic analysis constructed in our study more clearly, the setting logic of customer concentration effect on risk-taking is shown in Fig 1.

## 3. Data and methodology

### 3.1 Data and sample

Our primary sample includes data of all the listed firms on the Shanghai Stock Exchange and Shenzhen Stock Exchange from 2008 to 2018. Moreover, the financial, corporate governance, and supply chain data of the listed firms are collected from the CSMAR database. From the detailed data on the "top 5 major customers and suppliers: listed in the CSMAR database, we manually collected relevant supply chain information of the listed firms from their financial statement annotations, such as ranking order, the names of their top five suppliers and customers, as well as the purchase and sales amounts in their annual reports. In China, the supply chain information of the listed firms is from voluntary disclosures. According to researchers of corporate risk-taking [18, 21, 43] samples were screened as follows. (1) Financial firms and special treatment firms are omitted because of the industry and financial status. (2) firms that went public that year and observations without financial or customers' information are excluded. (3) The variables are winsorized at both the top and bottom 1% for each year. After the screening process, we obtain our final sample of 2,555 firms with 12,365 firm-year observations for analysis purposes.

### 3.2 Dependent variables

Motivated by previous studies [11, 12, 15, 16, 19], we consider four measures of risk-taking. Accounting performance volatility proxies for risk-taking are commonly used in the related topic: our primary measure of corporate risk-taking is Risk1—the volatility of a firm's quarterly return on asset (ROA), the ratio of operating income to total assets, based on the previous 5 years. From the study of Boubakri [18], we use Risk2—the difference between the maximum and minimum value of ROA over the previous 5 years—to proxy risk-taking. In addition, we estimate both Risk1 and Risk2 by using quarterly data over 3-year overlapping periods in our main tests, which refers to the other measures, Risk3 and Risk4. These measures are the outcomes of corporate risk-taking. The specific calculation formula of Risk1 is as following:

$$\text{RiskTaking}_i = \sqrt{\frac{1}{N-1}\sum_{n=1}^{N}\left(\text{ADJ}_{\text{Roain}} - \frac{1}{N}\sum_{n=1}^{N}\text{ADJ}_{\text{Roain}}\right)^2} \mid N = 5 \tag{1}$$

To simulate the riskiness of corporate policies, we consider the riskiness of both the investment and financial policies. In terms of investment policies, we adopt two measures of the riskiness of firm investment policies based on prior studies: R&D and capital expenditures [16, 44–46]. Corporate innovation input is regarded as a risky investment. For financial policies, Leverage (Lev) and Working Capital (Nwc) are commonly used to measure riskiness of a firm's financial policies [15, 16, 46]. Further higher leverage might lead to greater (negative) impact on a firm's net profitability. Measuring the riskiness of corporate policies is helpful for explaining how customer concentration affects corporate policies and captures the overall risk taken by the firm.

### 3.3 Independent variables

Based on related research [2, 4–6, 24], we use two variables to measure customer concentration. One measure is the customer concentration variable CustomerHHI, which is the sum of the squared percentage sales for each major customer, corresponding to the concentration based on the notion of a Herfindahl index of sales to major customers. The other measure is the total major customer sales ratio variable Customertop, which is the total percentage sales to all major customers.

### 3.4 Control variables

To better understand the impact of customer concentration on corporate decision-making, other relevant firm-specific variables should be controlled. We also control for other variables that affect a firm's risky behaviors, based on prior literature [15, 16, 21]. The following are the control variables: firm size Size, sales growth Grow, growth opportunities Q, firm fixed assets PPE, ownership structure Top1, and corporate nature Soe. Definitions of the variables are provided in the Table 1.

### 3.5 Models

The main independent variable of our analysis is customer concentration measured by CustomerHHI (Customertop). We also verify whether a nonlinear relationship exists between risk-taking, corporate policies and firm performance of a firm and the customer-supplier relationship, by including the squared term of CustomerHHI$^2$ (Customertop$^2$). We employ the cluster standard errors regression analysis and use the baseline model as follows:

$$Y_{i,t} = \beta_1 + \beta_2 \text{CustomerHHI}_{i,t} + \beta_3 \text{CustomerHHI2}_{i,t}^{\,2} + \beta_4 \text{Control}_{i,t} + \text{Industry} + \text{Year} + \varepsilon_{i,\,t} \tag{2}$$

**Table 1. Variable description.**

| Variable | Definition |
|---|---|
| Risk1 | The volatility of a firm's quarterly return on asset, constructed over the previous 5 years |
| Risk2 | Max (Roai,t)-Min(Roai,t) over the previous 5 years |
| Risk3 | The volatility of a firm's quarterly return on asset, constructed over the previous 3 years |
| Risk4 | Max (Roai,t)-Min(Roai,t) over the previous3 years |
| CustomeHHI | The notion of a Herfindahl index of sales to large customers, following Patatoukas (2012) |
| Customertop | The sum of firm's percentage sales made to its major customers (i.e., sales to a major customer / firm sales) |
| R&D | The ratio of research and development expenses to total assets |
| Cap | The ratio of capital expenditure to total assets |
| Lev | The ratio of total debt to total assets |
| Nwc | Current assets minus current liabilities, scaled by total assets, |
| Size | The natural log of total assets |
| Return | Stock return over the fiscal year t |
| Grow | Sales growth ratio |
| PPE | The ratio of fixed assets to total assets |
| Q | (Book value of debt+market value of equity)/book value of assets |
| Top1 | Share ratio of the largest shareholder |
| Soe | 1 if the firm is state-owned and 0 otherwise. |
| Dual | 1 if the CEO is also the chairman of the board, and 0 otherwise |
| Age | Ln (1+age of CEO) |
| Tenure | Ln (1+number of years since a CEO took office at that company) |
| Customersize | The firm's percentage sales to major customers weighted by the size of those customers, following Campello and Gao (2017), |
| VOL_daily | Volatility of daily stock returns for each firm year |
| VOL_month | Volatility of monthly stock returns for each firm year |
| ROA | Net profit/assets |

$Y_{i,t}$ represents our dependent variables, which are $Risk_{i,t}$, $Policies_{i,t}$ and $Performance_{i,t}$ respectively. Where $CustomerHHI_{i,t}$ represents customer concentration, $Control_{i,t}$ represents the firm-specific variables which has been mentioned. Similarly, the effects of Industry and Year should be controlled. We first calculate the effect of customer concentration on firm's risk-taking. We then calculate the effect of customer concentration on corporate policies. Finally, we calculate the effect of customer concentration on firm's financial performance using the equation.

## 4. Empirical results

### 4.1 Summary statistics

Table 2 presents the summary statistics for each of the main variables. Panel A shows the descriptive statistics for risk-taking measures and customer concentration measures in our panel. For the firm's risk-taking measures, the listed firms have an average of 0.043 for Risk1, a mean of 0.101 for Risk2. Further, the mean values of Risk3 and Risk4 are 0.035 and 0.147, with a standard deviation of 0.058 and 0.238, respectively. For customer concentration measures, the average customer concentration ratio CustomerHHI and the average percentage sales to all major customers Customertop are 5.1% and 13.6%, respectively. The R&D of the listed firms have an average level of 0.023, and Cap has a mean of 0.275. These data suggest that most of the listed firms in the sample have invested in R&D. The financial policy measure Lev has a

Table 2. Summary statistics of the main variables.

| Variable | Mean | Sd | Min | p50 | Max | N |
|---|---|---|---|---|---|---|
| Risk1 | 0.043 | 0.084 | 0.000 | 0.018 | 1.009 | 12,365 |
| Risk2 | 0.101 | 0.197 | 0.000 | 0.042 | 2.025 | 12,365 |
| Risk3 | 0.035 | 0.058 | 0.000 | 0.017 | 0.668 | 12,365 |
| Risk4 | 0.147 | 0.238 | 0.000 | 0.070 | 2.037 | 12,365 |
| CustomerHHI | 0.051 | 0.097 | 0.000 | 0.015 | 0.575 | 12,365 |
| Customertop | 0.136 | 0.147 | 0.004 | 0.085 | 0.837 | 12,365 |
| R&D | 0.023 | 0.032 | 0.000 | 0.008 | 0.164 | 12,365 |
| Cap | 0.275 | 0.188 | 0.002 | 0.241 | 0.828 | 12,365 |
| Lev | 0.438 | 0.218 | 0.048 | 0.431 | 0.990 | 12,365 |
| Nwc | 0.442 | 0.256 | -0.164 | 0.458 | 0.916 | 12,365 |
| Size | 21.917 | 1.226 | 18.883 | 21.770 | 25.621 | 12,365 |
| Return | 0.285 | 0.680 | -0.607 | 0.096 | 3.033 | 12,365 |
| Grow | 0.480 | 1.533 | -0.825 | 0.133 | 12.036 | 12,365 |
| PPE | 0.232 | 0.166 | 0.002 | 0.199 | 0.751 | 12,365 |
| Q | 2.106 | 1.577 | 0.525 | 1.603 | 10.172 | 12,365 |
| Top1 | 0.354 | 0.153 | 0.036 | 0.333 | 0.900 | 12,365 |
| Soe | 0.503 | 0.500 | 0.000 | 1.000 | 1.000 | 12,365 |
| ROA | 0.038 | 0.057 | -0.292 | 0.035 | 0.209 | 12,365 |

Note: This table reports summary statistics of the main variables used in the regression analysis, including firm risk-taking measures, the proxies of customer concentration and several control variables.

mean of 0.438 and a standard deviation of 0.218. Meanwhile, the mean of Nwc is approximately 0.442, with a standard deviation of 2.256. Panel B presents descriptive statistics for firms' characteristics in our panel. The average firm has the total asset level of 21.917. Return has an average (median) of 0.285 (0.680), and the average firm growth, Grow, is about 48%. The average fixed-asset ratio PPE is 0.232, while Q has a mean of 2.106, with a standard deviation of 1.577. In addition, the listed firms have an average level of Top1 of 35.4%, and 50.3% of these firms are stated owned.

## 4.2 Effect of customer concentration on corporate risk-taking

To examine the role of major customers in corporate governance, we focus on corporate risk-taking behavior and customer concentration. As the dependent variable is corporate risk-taking, we use four measures, as in prior research [15, 16, 19].

Table 3 presents our main results estimated using Eq (1). Columns 1–8 of the table display the results of regression tests of customer concentration on corporate risk-taking. The key variable CustomerHHI is positively and significantly related to risk-taking. The coefficients on CustomerHHI$^2$ are negative, which are significant at a 1% level, suggesting the nonlinear relationship between customer concentration and corporate risk-taking. Column (1) shows that the coefficient of CustomerHHI is positive and significant ($\beta$ = 0.153, t = 6.825). This suggests that major customers have a generally positive effect on corporate risk-taking. Such a positive effect is consistent with the certification hypothesis of customer concentration. Several researchers find that stable relationships with customers help suppliers achieve improved inventory management, greater operational efficiency, and new product success, enabling firms to allocate additional resources for value creation [3, 24]. Thus, suppliers benefit from reducing information asymmetry and effective access to external resources required and are

**Table 3. Effects of customer concentration on corporate risk-taking.**

|  | Risk1 | Risk2 | Risk3 | Risk4 | Risk1 | Risk2 | Risk3 | Risk4 |
|---|---|---|---|---|---|---|---|---|
| CustomerHHI | 0.153*** | 0.405*** | 0.116*** | 0.493*** |  |  |  |  |
|  | (6.825) | (6.287) | (7.476) | (6.712) |  |  |  |  |
| CustomerHHI$^2$ | -0.176*** | -0.444*** | -0.127*** | -0.492*** |  |  |  |  |
|  | (-3.389) | (-2.827) | (-3.498) | (-2.714) |  |  |  |  |
| Customertop |  |  |  |  | 0.097*** | 0.255*** | 0.076*** | 0.326*** |
|  |  |  |  |  | (6.685) | (6.122) | (7.421) | (6.684) |
| Customertop$^2$ |  |  |  |  | -0.078*** | -0.190** | -0.059*** | -0.232** |
|  |  |  |  |  | (-2.950) | (-2.380) | (-3.151) | (-2.495) |
| Size | -0.010*** | -0.025*** | -0.007*** | -0.031*** | -0.009*** | -0.024*** | -0.007*** | -0.030*** |
|  | (-13.144) | (-12.633) | (-13.938) | (-13.551) | (-12.594) | (-12.105) | (-13.281) | (-12.883) |
| Lev | 0.040*** | 0.101*** | 0.026*** | 0.109*** | 0.039*** | 0.100*** | 0.026*** | 0.108*** |
|  | (8.021) | (7.362) | (7.749) | (7.073) | (7.966) | (7.312) | (7.687) | (7.015) |
| Return | -0.009*** | -0.024*** | -0.007*** | -0.028*** | -0.009*** | -0.024*** | -0.007*** | -0.028*** |
|  | (-6.258) | (-6.209) | (-6.468) | (-6.186) | (-6.331) | (-6.276) | (-6.552) | (-6.264) |
| Grow | 0.003*** | 0.007*** | 0.002*** | 0.008*** | 0.003*** | 0.007*** | 0.002*** | 0.008*** |
|  | (4.148) | (3.566) | (3.917) | (3.456) | (4.189) | (3.608) | (3.969) | (3.503) |
| PPE | 0.017*** | 0.045*** | 0.009*** | 0.049*** | 0.018*** | 0.047*** | 0.010*** | 0.053*** |
|  | (3.816) | (3.539) | (2.860) | (3.266) | (4.002) | (3.715) | (3.080) | (3.483) |
| Q | 0.008*** | 0.020*** | 0.006*** | 0.027*** | 0.008*** | 0.021*** | 0.006*** | 0.027*** |
|  | (10.645) | (10.041) | (12.310) | (11.800) | (10.771) | (10.150) | (12.465) | (11.944) |
| Top1 | -0.042*** | -0.116*** | -0.035*** | -0.160*** | -0.042*** | -0.115*** | -0.034*** | -0.159*** |
|  | (-10.466) | (-10.338) | (-12.346) | (-12.170) | (-10.393) | (-10.269) | (-12.270) | (-12.101) |
| Soe | -0.005*** | -0.013*** | -0.004*** | -0.021*** | -0.005*** | -0.014*** | -0.004*** | -0.022*** |
|  | (-2.866) | (-2.955) | (-3.764) | (-3.962) | (-3.047) | (-3.133) | (-3.977) | (-4.180) |
| Cons | 0.237*** | 0.527*** | 0.179*** | 0.687*** | 0.226*** | 0.498*** | 0.170*** | 0.647*** |
|  | (14.142) | (12.450) | (14.936) | (13.590) | (13.282) | (11.529) | (13.977) | (12.517) |
| Year | Yes | Yes | Yes | Yes | Yes | Yes | Yes | Yes |
| Industry | Yes | Yes | Yes | Yes | Yes | Yes | Yes | Yes |
| N | 12,365 | 12,365 | 12,365 | 12,365 | 12,365 | 12,365 | 12,365 | 12,365 |
| Adj. R2 | 0.131 | 0.123 | 0.152 | 0.148 | 0.131 | 0.123 | 0.153 | 0.149 |
| F-test | 67.558*** | 28.020*** | 33.532*** | 30.481*** | 29.275*** | 21.021 *** | 59.654*** | 49.950*** |

Note: This table reports the OLS estimation results. The dependent variables in regressions are the corporate risk-taking Risk1, Risk2, Risk3 and Risk4. The independent variables are customer concentration measures CustomeHHI and Customertop. The standard errors are given in parentheses. Significance levels

***p < 0.01

** p < 0.05, and

* p < 0.1.

more likely to choose risky investment projects, leading to higher risk-taking level. Nevertheless, CustomerHHI$^2$ takes on a negative and significant coefficient (β = -0.1760.153, t = -3.389). As discussed previously, if customer concentration is remarkably high, customer bargaining power is enhanced. Considering higher concentrated credit risk and switch costs, suppliers avoid risks and reduce investments on risky projects [4, 9, 10, 34].

The control variables in these regressions are the firm characteristics that are typically used as determinants of risk-taking. We find that risk-taking is negatively related to firm size, stock return, firm age, and shareholder equity ratio and it is positively related to leverage, sales growth, fixed asset ratio, and investment opportunities proxied by Tobin's q. From our results,

we suggest that the effect of customer concentration in China perfectly resembles an inverted "U," which is consistent with the hypotheses about major customer's function with different levels of their customer concentration (certification hypothesis and concentrated credit risk hypothesis). Firms with several major customers increase risk-taking, but substantially concentrated customers have the opposite effect, as we have discussed in the Introduction. We confirm that an inverse U-shaped curve between Grow and corporate risk-taking exists.

### 4.3 Customer concentration and corporate policies

In this section, we test the channels that major customers affect and subsequently investigate the relationship between customer concentration and corporate policies. We examine the association between customer concentration and investment policies as proxied by R&D and capital expenditures.

Table 4 shows the results regarding the relationship between customer concentration and corporate investment policies. The dependent variable in columns 1–2 is R&D. Prior research [47] suggests that R&D expenditures are riskier than other investment choices. We find a positive and significant effect of CustomerHHI on risk-taking, and the coefficients on CustomerHHI2 are negative, which are significant at a 1% level. We find similar results when we focus on the effect of CustomerHHI on capital expenditures in columns 3–4, suggesting a nonlinear relationship between customer concentration and corporate investment policies.

In addition, we analyze the relationship between customer concentration and corporate financial policies. Increased financial policy risks are reflected by holding fewer liquid assets or by increasing the firm's debt burden [41]. Thus, we expect a positive relationship between customer concentration and financial leverage and a negative association between customer concentration and asset liquidity (working capital). Consequently, we find that a U-shaped relationship exists between customer concentration and Lev in columns 5–6 of Table 2. Columns (7)–(8) present estimation results of model (1), where Nwc is used. We further discover that the relationship between customer concentration and net working capital is U-shaped.

The results depicted in Table 2 combined demonstrate that major customers increase or decrease corporate risk-taking and financial policies based on the level of customer concentration, which is consistent with our prior findings.

### 4.4 Customer concentration and firm performance

The characteristics of customer-supplier relationship can influence the financial decision-making, and then affect the value of the firm. In this section we examine the link between customer concentration and firm performance. Since there is an inverse U-shaped relationship between customer concentration and risk-taking, we predict a U-shaped relation between customer concentration and firm performance. Following methodology developed by prior research [39, 40] we use measures of firm performance as the dependent variables, The dependent variable in regressions (1)-(2) is financial performance ROA. The dependent variable in regressions (3)-(4) is market performance TobinQ.

Table 5 report the effects of customer concentration and firm performance. In columns 1–2, the coefficients of CustomerHHI are positive, and the coefficients on CustomerHHI$^2$ are negative, which are significant at a 1% level. In columns 3–4, the coefficients of Customertop are also positive, and the coefficients on Customertop$^2$ are negative, thus suggesting that there is an inverse U-shaped curve between Grow and corporate performance. As mentioned earlier, this finding appears to have a rational explanation. Firms with low customer concentration are more likely to take higher risks for supply chain stability, which could benefit the listed firms

**Table 4. Effects of customer concentration on corporate policies.**

|  | R&D | | Cap | | Lev | | Nwc | |
|---|---|---|---|---|---|---|---|---|
| CustomerHHI | 0.016* | | 0.060*** | | 0.190*** | | -0.277*** | |
|  | (1.718) | | (2.696) | | (4.944) | | (-4.381) | |
| CustomerHHI$^2$ | -0.069** | | -0.221*** | | -0.216** | | 0.721*** | |
|  | (-2.524) | | (-3.539) | | (-2.554) | | (3.904) | |
| Customertop | | 0.026*** | | 0.052*** | | 0.200*** | | -0.265*** |
|  | | (4.231) | | (3.703) | | (7.408) | | (-6.614) |
| Customertop$^2$ | | -0.053*** | | -0.112*** | | -0.217*** | | 0.435*** |
|  | | (-4.656) | | (-4.463) | | (-4.754) | | (5.794) |
| Size | 0.000 | 0.000 | 0.008*** | 0.008*** | 0.040*** | 0.041*** | -0.032*** | -0.033*** |
|  | (0.058) | (0.452) | (12.020) | (12.031) | (29.816) | (30.512) | (-18.006) | (-18.467) |
| Return | 0.004*** | 0.004*** | -0.004*** | -0.004*** | 0.006** | 0.006** | -0.001 | -0.001 |
|  | (9.620) | (9.575) | (-3.770) | (-3.841) | (2.136) | (2.052) | (-0.398) | (-0.303) |
| Grow | -0.001*** | -0.001*** | -0.001*** | -0.001*** | -0.000 | -0.000 | 0.004*** | 0.004*** |
|  | (-3.805) | (-3.789) | (-4.576) | (-4.529) | (-0.024) | (-0.005) | (3.190) | (3.156) |
| PPE | -0.016*** | -0.016*** | 1.010*** | 1.010*** | 0.289*** | 0.292*** | -0.949*** | -0.951*** |
|  | (-10.108) | (-10.209) | (250.079) | (250.214) | (28.789) | (29.063) | (-81.710) | (-81.722) |
| Q | 0.000** | 0.000** | -0.001*** | -0.001*** | -0.003*** | -0.003*** | -0.000 | -0.001 |
|  | (2.433) | (2.463) | (-3.234) | (-3.150) | (-2.965) | (-2.835) | (-0.231) | (-0.332) |
| Top1 | -0.005*** | -0.005*** | -0.002 | -0.002 | -0.095*** | -0.095*** | 0.178*** | 0.177*** |
|  | (-2.627) | (-2.600) | (-0.542) | (-0.461) | (-11.366) | (-11.305) | (16.308) | (16.200) |
| Lev | -0.038*** | -0.038*** | 0.006* | 0.005* | | | | |
|  | (-28.697) | (-28.698) | (1.944) | (1.879) | | | | |
| Soe | -0.018*** | -0.018*** | -0.005*** | -0.005*** | 0.002 | 0.002 | -0.011*** | -0.011*** |
|  | (-31.295) | (-31.447) | (-3.609) | (-3.626) | (0.752) | (0.472) | (-2.801) | (-2.629) |
| Cons | 0.051*** | 0.047*** | -0.040* | -0.045** | -0.536*** | -0.570*** | 1.145*** | 1.179*** |
|  | (8.976) | (8.211) | (-1.803) | (-1.999) | (-11.542) | (-12.209) | (21.869) | (22.313) |
| Year | Yes | Yes | Yes | Yes | Yes | Yes | Yes | Yes |
| Industry | Yes | Yes | Yes | Yes | Yes | Yes | Yes | Yes |
| N | 12,365 | 12,365 | 12,365 | 12,365 | 12,365 | 12,365 | 12,365 | 12,365 |
| Adj. R2 | 0.218 | 0.219 | 0.895 | 0.895 | 0.266 | 0.268 | 0.523 | 0.524 |
| F-test | 345.474*** | 346.713*** | 3569.365*** | 3560.056*** | 136.572*** | 159.677*** | 399.762*** | 403.068*** |

Note: This table reports effect of customer concentration on corporate policies. The dependent variables in regressions are the R&D, capital expenditures Cap, leverage Lev and Working Capital Nwc. The independent variables are customer concentration measures CustomeHHI and Customertop. The standard errors are given in parentheses. Significance levels

***$p < 0.01$

** $p < 0.05$, and

* $p < 0.1$.

for long-run development. While those with high customer concentration are more likely to take less risks, resulting in lower level of financial performance.

The goal of this article is to examine the impact of customer concentration on risk-taking and corporate governance in emerging markets. Herein, we empirically investigate the effect of customer concentration on corporate risk-taking using a sample of Chinese firms. Specifically, we used the proxy of customer concentration based on the studies of Cai and Zhu [2] and Dhaliwal et al [4]. First, there is an inverse U-shaped curve exists between customer concentration and corporate risk-taking. As the results shows that hosting firms with low

**Table 5. Effect of customer concentration on corporate performance.**

| | ROA | | TobinQ | |
|---|---|---|---|---|
| | **(1)** | **(2)** | **(3)** | **(4)** |
| CustomerHHI | 0.058*** | | 0.299* | |
| | (3.964) | | (1.686) | |
| CustomerHHI$^2$ | -0.117*** | | -0.778** | |
| | (-3.545) | | (-2.262) | |
| Customertop | | 0.055*** | | 0.204* |
| | | (5.502) | | (1.701) |
| Customertop$^2$ | | -0.077*** | | -0.374** |
| | | (-4.418) | | (-2.052) |
| Size | 0.013*** | 0.013*** | -0.417*** | -0.417*** |
| | (21.150) | (20.674) | (-31.883) | (-31.940) |
| Lev | -0.128*** | -0.128*** | 0.648*** | 0.648*** |
| | (-32.524) | (-32.485) | (14.571) | (14.563) |
| Return | 0.015*** | 0.015*** | 0.261*** | 0.261*** |
| | (14.488) | (14.530) | (37.294) | (37.356) |
| Grow | 0.002*** | 0.002*** | -0.006** | -0.006** |
| | (4.836) | (4.842) | (-2.325) | (-2.338) |
| PPE | -0.053*** | -0.054*** | 0.149** | 0.150** |
| | (-15.067) | (-15.178) | (2.496) | (2.505) |
| Top1 | 0.027*** | 0.027*** | -0.767*** | -0.769*** |
| | (8.903) | (8.869) | (-9.006) | (-9.012) |
| Soe | -0.005*** | -0.005*** | 0.228*** | 0.229*** |
| | (-4.491) | (-4.360) | (13.130) | (13.153) |
| Cons | -0.178*** | -0.170*** | 8.464*** | 8.468*** |
| | (-13.055) | (-12.313) | (24.481) | (24.503) |
| Year | Yes | Yes | Yes | Yes |
| Industry | Yes | Yes | Yes | Yes |
| N | 10,754 | 10,754 | 10,754 | 10,754 |
| Adj. R2 | 0.260 | 0.261 | 0.603 | 0.603 |
| F-test | 59.679*** | 59.967*** | 183.496*** | 183.931 *** |

Note: This table reports the effect of customer concentration on corporate performance. The dependent variables in regressions are the firm performance ROA and TobinQ. The independent variables are customer concentration measures CustomeHHI and Customertop. The standard errors are given in parentheses. Significance levels

***$p < 0.01$

** $p < 0.05$, and

* $p < 0.1$.

customer concentration are more likely to take higher risks for supply chain stability, while those with high customer concentration are more likely to take less risks. Second, we present the channels. According to the effects of customer concentration on corporate financial and investment policies, a prominent inverse-U shape relationship is observed between customer concentration and financial leverage, R&D investment and capital expenditures. whereas a U-shaped relationship is observed between customer concentration and asset liquidity. Finally, we provide sufficient support for the inverted U-shaped relationship between customer concentration and firm performance. Our results suggest that having balanced and relatively high customer concentration benefit firm in reducing information asymmetry and effectively

undertake risky investment projects. We empirically document that an inverse U-shaped curve exists between customer concentration and corporate risk-taking.

## 5. Endogeneity tests and robustness checks

### 5.1 Endogeneity tests

The results presented suggest that major customers significantly affect the volatility of earnings as well as the risk of corporate policies. However, these inverse U-shaped relationships might be contaminated by the potential endogeneity. To address the endogeneity problems, we employ change model framework in accordance to the study of Patatoukas [24]. First, we calculate the difference between CustomerHHIt and CustomerHHIt-1. Second, the calculated ΔCustomerHHIt is divided into two groups according to the positive and negative values. Third, we take the negative ΔCustomerHHIt as the absolute value │ΔCustomerHHIt│. Finally, the square terms of the positive ΔCustomerHHIt and │ΔCustomerHHIt│ are used. Among them, positive ΔCustomerHHIt represents the increased customer concentration in this year, and │ΔCustomerHHIt│ indicates the reduction of enterprise customer concentration in that year.

Table 6 reports the results of the change model. Columns 1–2 of Panel A show that the coefficients on ΔCustomerHHIt are positive and significant and the coefficients on ΔCustomerHHI$^2$ are negative and significant. This suggests that when customer concentration increases, the level of corporate risk-taking will also increase. However, when customer concentration reaches a certain level and continuously increases, the level of corporate risk-taking will decrease. In columns 3–4, when customer concentration increases, the level of corporate risk-taking will increase. However, when customer concentration reaches a certain level and continuously reduces, the level of corporate risk-taking will decrease. Panel B shows the results of the relationship between Customertop and firm's risk-taking. The results reconfirm our initial findings of inverse U-shaped relationship between the customer concentration and risk-taking.

### 5.2 Robustness checks

In this section, we examine whether our primary results are robust to applying alternative measures and modifying the control variables. However, it could be more likely that major customers happen to be larger firms; hence, our measure of customer concentration could be correlated with the firm size of customers. We address this possibility by considering the firm size of major customers. Based on the study of Campello and Gao [5], we reexamined the results by conducting alternative measures of customer concentration CustomerSize, which is defined as the firm's percentage sales to major customers weighted by the firm size of these customers. Table 7 shows the relationship between customer concentration and corporate risk-taking in Section 3.1 using the alternative customer concentration measures. In column 1, the coefficients CustomerSize and CustomerSize$^2$ are 0.494 (t = 8.702) and -1.276 (t = -4.190), respectively, which are significant. This indicates the nonlinear relationship between customer concentration and corporate risk-taking.

Furthermore, we have alternative methods for identifying alternative measures of corporate risk-taking and for running the panel regression. Similar to previous risk-taking studies for firms [16, 48], we used VOL (return_daily) and VOL (return_month) as the dependent variable—volatility—of daily or monthly stock returns, respectively, formulated over the window from t + 1 to t + 3. Table 8 shows the significant inverse U-shaped relationship between customer concentration and firms' level of risk-taking. We found that the mean results do not change, and all the estimated coefficients on the square terms of customer concentration

**Table 6. Estimations results of Changes Model analysis.**

| | ΔRisk1 | ΔRisk2 | ΔRisk1 | ΔRisk2 |
|---|---|---|---|---|
| **Panel A. Customer concentration** = *CustomerHHI* | | | | |
| ΔCustomerHHI | 0.072** | 0.338*** | | |
| | (1.975) | (3.544) | | |
| \|ΔCustomerHHI\| | | | 0.485*** | 1.358*** |
| | | | (4.600) | (4.857) |
| ΔCustomerHHI$^2$ | -0.872* | -3.124** | -2.503** | -7.105*** |
| | (-1.666) | (-2.280) | (-2.508) | (-2.683) |
| ΔSize | -0.017*** | -0.043*** | -0.014** | -0.042** |
| | (-4.624) | (-4.415) | (-2.143) | (-2.372) |
| ΔReturn | 0.000 | -0.000 | 0.002 | 0.008 |
| | (0.269) | (-0.094) | (0.861) | (1.143) |
| ΔLev | 0.016 | 0.037 | -0.007 | -0.016 |
| | (1.517) | (1.317) | (-0.397) | (-0.344) |
| ΔGrow | 0.001** | 0.003** | 0.002*** | 0.005** |
| | (2.517) | (2.140) | (2.681) | (2.450) |
| ΔPPE | -0.003 | 0.035 | -0.040* | -0.105* |
| | (-0.217) | (1.087) | (-1.773) | (-1.756) |
| ΔQ | -0.003*** | -0.008*** | -0.000 | -0.001 |
| | (-2.663) | (-2.652) | (-0.103) | (-0.196) |
| ΔTop1 | 0.004 | -0.018 | 0.049* | 0.152** |
| | (0.210) | (-0.327) | (1.686) | (1.982) |
| Soe | -0.003 | -0.009 | -0.002 | -0.005 |
| | (-1.576) | (-1.539) | (-0.212) | (-0.281) |
| Cons | 0.008 | 0.036 | -0.048 | -0.147* |
| | (0.292) | (0.498) | (-1.496) | (-1.737) |
| Year | Yes | Yes | Yes | Yes |
| Industry | Yes | Yes | Yes | Yes |
| N | 2,439 | 2,439 | 1,688 | 1,688 |
| Adj. R2 | 0.030 | 0.053 | 0.189 | 0.207 |
| F-test | 1.861*** | 3.379*** | 4.474*** | 5.009*** |
| **Panel B. Customer concentration** = *Customertop* | | | | |
| | ΔRisk1 | ΔRisk2 | ΔRisk1 | ΔRisk2 |
| ΔCustomertop | -0.033* | 0.317*** | | |
| | (-1.925) | (3.827) | | |
| \|ΔCustomertop\| | | | 0.371*** | 0.995*** |
| | | | (3.678) | (3.692) |
| ΔCustomertop$^2$ | 0.260** | -1.266** | -1.210** | -3.225** |
| | (2.313) | (-2.356) | (-2.096) | (-2.093) |
| ΔSize | 0.003 | -0.044*** | -0.010 | -0.037** |
| | (1.366) | (-4.275) | (-1.455) | (-2.079) |
| ΔReturn | -0.000 | 0.000 | -0.000 | 0.000 |
| | (-0.220) | (0.064) | (-0.125) | (0.045) |
| ΔLev | 0.004 | 0.038 | -0.014 | -0.038 |
| | (0.743) | (1.331) | (-0.826) | (-0.864) |
| ΔGrow | -0.000 | 0.004*** | 0.003*** | 0.007*** |
| | (-0.483) | (2.888) | (3.222) | (2.995) |
| ΔPPE | 0.004 | -0.019 | -0.029 | -0.097 |

(*Continued*)

**Table 6.** (Continued)

|  | (0.494) | (-0.561) | (-1.276) | (-1.576) |
|---|---|---|---|---|
| ΔQ | 0.001* | -0.007** | 0.002 | 0.006 |
|  | (1.684) | (-2.556) | (1.147) | (1.213) |
| ΔTop1 | -0.003 | -0.054 | 0.053* | 0.164** |
|  | (-0.280) | (-1.023) | (1.783) | (2.064) |
| Soe | -0.001 | 0.005 | -0.001 | -0.008 |
|  | (-1.216) | (0.809) | (-0.159) | (-0.371) |
| Cons | 0.004 | -0.015 | -0.046* | -0.144* |
|  | (0.248) | (-0.212) | (-1.648) | (-1.933) |
| Year | Yes | Yes | Yes | Yes |
| Industry | Yes | Yes | Yes | Yes |
| N | 2,421 | 2,421 | 1,648 | 1,648 |
| Adj. R2 | 0.041 | 0.044 | 0.320 | 0.335 |
| F-test | 2.523*** | 2.723*** | 9.211*** | 9.857*** |

Note: This table presents estimations results of Changes Model analysis. The standard errors are given in parentheses. Significance levels

***$p < 0.01$

** $p < 0.05$, and

* $p < 0.1$.

measures CustomerHHI$^2$ and Customertop$^2$ are negative, suggesting that major customers affect the riskiness of a firm.

In a robustness test, we also included CEO characteristics as additional controls to include the possibility of CEOs engaging in different levels of risk-taking. We suggested measures of CEO characteristics, including CEO duality (Dual is equal to 1 if the CEO serves as board chair, or otherwise, 0), CEO age (Age, the natural logarithm of one plus the age of CEO), and CEO tenure (Tenure, the natural logarithm of one plus the number of years since a CEO took

**Table 7. Robustness test 1: Alternative measures of customer concentration.**

|  | Risk1 | Risk2 | Risk3 | Risk4 |
|---|---|---|---|---|
| CustomerSize | 0.494*** | 1.321*** | 0.371*** | 1.631*** |
|  | (8.702) | (8.096) | (9.383) | (8.689) |
| CustomerSize$^2$ | -1.276*** | -3.203*** | -0.915*** | -3.696*** |
|  | (-4.190) | (-3.477) | (-4.319) | (-3.474) |
| Control | Yes | Yes | Yes | Yes |
| Cons | 0.211*** | 0.455*** | 0.159*** | 0.596*** |
|  | (12.345) | (10.526) | (13.116) | (11.536) |
| Year | Yes | Yes | Yes | Yes |
| Industry | Yes | Yes | Yes | Yes |
| N | 12,365 | 12,365 | 12,365 | 12,365 |
| Adj. R2 | 0.134 | 0.127 | 0.155 | 0.152 |
| F-test | 112.543*** | 36.269*** | 32.150*** | 29.177*** |

Note: This table shows robustness test using the alternative customer concentration measures. The standard errors are given in parentheses. Significance levels

***$p < 0.01$

** $p < 0.05$, and

* $p < 0.1$.

**Table 8. Robustness test 2: Alternative measures of corporate risk-taking.**

| | VOL (return_daily) | | VOL (return_month) | |
|---|---|---|---|---|
| CustomerHHI | 0.159*** | | 0.069*** | |
| | (3.764) | | (2.708) | |
| CustomerHHI$^2$ | -0.313*** | | -0.167*** | |
| | (-3.788) | | (-3.316) | |
| Customertop | | 0.113*** | | 0.058*** |
| | | (4.224) | | (3.750) |
| Customertop$^2$ | | -0.145*** | | -0.089*** |
| | | (-3.797) | | (-4.046) |
| Control | Yes | Yes | Yes | Yes |
| Cons | 0.760*** | 0.747*** | 0.802*** | 0.795*** |
| | (20.748) | (20.182) | (35.465) | (34.798) |
| Year | Yes | Yes | Yes | Yes |
| Industry | Yes | Yes | Yes | Yes |
| N | 12,365 | 12,365 | 12,365 | 12,365 |
| Adj. R2 | 0.502 | 0.502 | 0.678 | 0.678 |
| F-test | 156.124*** | 156.412*** | 485.327*** | 485.063*** |

Note: This table shows robustness test using the alternative measures of corporate risk-taking. The standard errors are given in parentheses. Significance levels ***$p < 0.01$, ** $p < 0.05$, and * $p < 0.1$.

office at that firm). The results of this robustness test are presented in Table 9. In summary, the main results are not affected by the inclusion of additional control variables.

To alleviate the concern that endogeneity is related to this research, we provide detailed approach in several ways. First, we use an change model approach, following Patatoukas [24] and reconfirm our initial findings of inverse U-shaped relationship between the customer concentration and risk-taking. Second, we examine whether our primary results are robust to applying alternative measures. We use the alternative customer concentration measures by considering the firm size of major customers [5], and by conducting alternative measures of customer risk-taking. Finally, this paper further mitigates the concern by modifying the measures of CEO characteristics as control variables. We find that the key findings that the inverse U-shaped impact of the customer concentration on corporate risk-taking still hold.

## 6. Conclusion

Understanding the role of customer concentration in corporate governance is important because the customer, the most valuable resource, is the foundation for a firm's survival. Previous research shows the effect of customer concentration on corporate decision-making, but the findings on the role of major customers vary. In this study, we highlight the importance of customer concentration for explaining risk-taking choices, corporate policies and firm performance based on data from the Chinese stock market.

Based on a large growing number of the literature, this paper proposes two hypotheses (certification hypothesis and concentrated credit risk hypothesis) regarding the impact of customer concentration on corporate risk-taking using panel data from 2,555 firms with 12,365 firm-year observations from 2008 to 2018 and a spatial Durbin model to empirically test the research hypotheses. We handly collected supply chain data from the Financial Statement Annotations and financial accounting information from the China Stock Market & Accounting Research Database (CSMAR) databases and get the following findings. We investigated the

**Table 9. Robustness test 3: Additional control variables.**

|  | Risk1 | Risk2 | Risk3 | Risk4 |
|---|---|---|---|---|
| CustomerHHI | 0.180*** | 0.129*** | 0.495*** | 0.405*** |
|  | -6.535 | -6.834 | -6.743 | -6.303 |
| CustomerHHI$^2$ | -0.198*** | -0.125*** | -0.504*** | -0.451*** |
|  | (-3.016) | (-2.691) | (-2.780) | (-2.870) |
| Size | -0.010*** | -0.008*** | -0.031*** | -0.025*** |
|  | (-12.478) | (-13.268) | (-13.325) | (-12.479) |
| Return | -0.010*** | -0.007*** | -0.028*** | -0.024*** |
|  | (-6.152) | (-6.468) | (-6.223) | (-6.212) |
| Lev | 0.044*** | 0.028*** | 0.105*** | 0.098*** |
|  | -7.591 | -7.408 | -6.863 | -7.208 |
| Grow | 0.003*** | 0.002*** | 0.008*** | 0.007*** |
|  | -3.685 | -3.454 | -3.336 | -3.463 |
| PPE | 0.016*** | 0.008** | 0.049*** | 0.044*** |
|  | -2.924 | -2.116 | -3.23 | -3.459 |
| Q | 0.009*** | 0.007*** | 0.027*** | 0.020*** |
|  | -10.044 | -11.807 | -11.851 | -10.055 |
| Top1 | -0.049*** | -0.040*** | -0.163*** | -0.118*** |
|  | (-10.360) | (-12.345) | (-12.388) | (-10.498) |
| Soe | -0.006*** | -0.005*** | -0.023*** | -0.016*** |
|  | (-3.212) | (-3.810) | (-4.341) | (-3.465) |
| Dual | -0.004** | -0.001 | -0.005 | -0.008** |
|  | (-2.170) | (-1.100) | (-0.977) | (-1.977) |
| Age | 0.014** | 0.010** | 0.042** | 0.031** |
|  | -2.254 | -2.442 | -2.399 | -2.078 |
| Tenure | -0.005*** | -0.005*** | -0.018*** | -0.012*** |
|  | (-5.576) | (-7.167) | (-6.873) | (-5.297) |
| Cons | 0.210*** | 0.157*** | 0.562*** | 0.437*** |
|  | -7.486 | -8.057 | -7.021 | -6.516 |
| Year | Yes | Yes | Yes | Yes |
| Industry | Yes | Yes | Yes | Yes |
| N | 12,365 | 12,365 | 12,365 | 12,365 |
| Adj. R2 | 0.126 | 0.151 | 0.152 | 0.126 |
| F-test | 42.750*** | 31.989*** | 31.237*** | 23.990*** |

Note: This table shows robustness test by inclusion of additional control variables. The standard errors are given in parentheses. Significance levels

***p < 0.01

** p < 0.05, and

* p < 0.1.

relationship between customer concentration and a firm's risk-taking behavior and found that the relationship is not one directional.

The results show that firms have a higher level of risk-taking when their customer concentration is low. This is because they benefit from reducing information asymmetry and effective access to external resources required and are more likely to choose risky investment projects. However, if firms' customer concentration is high, they can avoid risk and reduce risk project investment owing to customers' bargaining power, higher concentrated credit risk, and switch costs. We empirically document that an inverse U-shaped curve exists between customer

concentration and corporate risk-taking. Furthermore, we showed that major customers cause corporate policy actions and increase or decrease riskiness of firm investment and financial policies based on the level of customer concentration. Given the inverse U-shaped effect of firms with customer concentration on corporate risk-taking, the major customer directly affects the consequence of corporate governance—firm performance. In addition, the empirical results do not stem from the endogeneity, robust to alternative measures of customer concentration and risk-taking and CEO-level control variables. We empirically document that an inverse U-shaped curve exists between customer concentration and corporate risk-taking.

In summary, our results highlight the importance of customer concentration in corporate risk-taking behavior, representing channels and economic consequence toward a better understanding of this important issue. The role of customer concentration could be treated as "double-edged sword" for the hosting firms. On the one hand, good cooperation among enterprise members in the supply chain and the sharing of technical resources can optimize resource allocation and improve the stability of supply chain relations. On the other hand, driven by the different goals, information asymmetry and benefit maximization among supply chain members, enterprises are at the risk of opportunistic behavior, which may lead to the "hold-up" problem. This paper finds that customer concentration can influence corporate governance and performance by affecting risk-taking. Our results suggest that having balanced and relatively high customer concentration benefit firm in reducing information asymmetry and effectively undertake risky investment projects. As a consequence, customer concentration should be maintained at a reasonable level and the results offer important insights on major customers and how they shape corporate decisions about risk and project investment in corporate governance. These points have important implications for policymaker and enterprises in emerging market country.

## Supporting information

**S1 File. Data.**
(DTA)

## Author Contributions

**Conceptualization:** Di Gao, Jiangming Ma.

**Data curation:** Di Gao.

**Formal analysis:** Di Gao, Jiangming Ma.

**Methodology:** Di Gao, Jiangming Ma.

**Visualization:** Di Gao.

**Writing – original draft:** Di Gao, Jiangming Ma.

**Writing – review & editing:** Di Gao, Jiangming Ma, Yiru Wang.

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
