## [Decision Letter · Decision Letter 0]

3 Sep 2021

PONE-D-21-18289Does the risk of major customer need to be balanced? The role of customer concentration in corporate governancePLOS ONE

Dear Dr. Gao,

Thank you for submitting your manuscript to PLOS ONE. After careful consideration, we feel that it has merit but does not fully meet PLOS ONE’s publication criteria as it currently stands. Therefore, we invite you to submit a revised version of the manuscript that addresses the points raised during the review process.

We look forward to receiving your revised manuscript.

Kind regards,

László Vasa, PhD

Academic Editor

PLOS ONE

A clean copy of the edited manuscript (uploaded as the new *manuscript* file).

“The authors declare no conflict of interest.”

Additional Editor Comments (if provided):

Reviewers' comments:

Reviewer's Responses to Questions

**Comments to the Author**

1. Is the manuscript technically sound, and do the data support the conclusions?

Reviewer #1: Partly

Reviewer #2: Yes

2. Has the statistical analysis been performed appropriately and rigorously? 

Reviewer #1: Yes

Reviewer #2: Yes

3. Have the authors made all data underlying the findings in their manuscript fully available?

Reviewer #1: Yes

Reviewer #2: Yes

4. Is the manuscript presented in an intelligible fashion and written in standard English?

Reviewer #1: Yes

Reviewer #2: Yes

5. Review Comments to the Author

Reviewer #1: Manuscript discusses interesting topic supported by statistical analyses, but needs improvement.

Abstract does not give the research goal directly. It should be reconstructed in my opinion. A clear, simple abstract would attract readers, in this form it is complicated and does not focus on the real goal of the authors.

Introduction is too long with its 5 pages. Again, a more simple structure would help. I suggest to visualize conceptual model of the research to clarify.

Literature review is good, I suggest to reconsider the hypotheses. Is it important to cut them into A and B parts? They focus on the same thing from the opposite angle.

Materials and methods: plain text is too much. Authors should try to visualize the content at least by using listed form.

Results: The content of S1 table should be inserted into the text. This table defines the terms which are used in the results chapter. Without that information the content is not clear.

The legend for the abbreviations shall be given below the tables. The format of the tables is not clear enough. The conclusions derived from the analyses should be highlighted at the end of the subchapters, at least in a new paragraph.

The asterisks ***, **, * are very confusing in the tables. Other indication should be used.

In the present form it is hard to find what was the aim for conducting the performed analyses. Again, I recommend to add a visualized conceptual model which may help to follow the process of the research, how the analyses lead us to the new results.

Conclusions: must be broadened. It says: "We empirically document that an inverse U-shaped curve exists between customer concentration and corporate risk-taking." Yes, but it is not highlighted in the results. Please show it.

It is seen that authors presented a complex research, but the way of presentation must be improved for acceptance.

Reviewer #2: The paper investigates the role of customer concentration on corporate governance and how the risks of main customers can influence the company.

The study is well written, its structure is perfect (except the keywords which are missing).

The abstract is well written, giving a brief highlight of the content in an excellent style.

The introduction explains the importance of the topic and introduces the importance and context, supported by some literature sources.

The literature review is well based, critical, and analytical enough; the hypotheses are developed in this framework too.

The methodology is clearly written, logical. Moreover, the authors selected the appropriate methodological toolset and applied them very well.

The methodology well supports the results; the conclusions are based on the results.

The language of the paper is scientific enough and well understandable.

6. PLOS authors have the option to publish the peer review history of their article (what does this mean?). If published, this will include your full peer review and any attached files.

Reviewer #1: No

Reviewer #2: No

---

## [Author Response · Author response to Decision Letter 0]

29 Sep 2021

Response to Reviewer 1

Title:Does the risk of major customer need to be balanced? The role of customer concentration in corporate governance

(PONE-D-21-18289)

We appreciate the comments and suggestions in your referee’s report, which have helped us to better position the paper. We have carefully revised the paper to incorporate your suggestions. We believe that the incorporation of your suggestions has made significant improvement to this paper. The revised part of the paper is in blue. Below please find our point-by-point responses, with the referees’ original comments shown in italics. And the revised manuscript are attached.

Reviewer #1: Manuscript discusses interesting topic supported by statistical analyses, but needs improvement.

1. Abstract does not give the research goal directly. It should be reconstructed in my opinion. A clear, simple abstract would attract readers, in this form it is complicated and does not focus on the real goal of the authors.

Response: 

We are very thankful for this comment. As for the comments, we are sorry that we do not give the research goal directly. In the rewrote Abstract and added more discussions of the research goal related to our topic in Introduction, Result and Conclusion. Please see revised Abstract, on page 1.

2.Introduction is too long with its 5 pages. Again, a more simple structure would help. I suggest to visualize conceptual model of the research to clarify.

Response: 

The referee’s comments are reasonable, and we revised the paper accordingly. 

First,we reorganized Introduction with 4 pages by abbreviating research contribution. Second, we deleted duplicate similar statements. 

Third, we added a new paragraph to explain the chapter arrangement and structure.

Fourth, we added a figure to clearly clarify the economic logic framework of our research in the revised manuscript, The economic channels or research logic frame is as follows:

Fig. 1. Logical diagram of customer concentration effect on firm risk-taking.

Please see revised Section 1, on page 2-5,9.

3.Literature review is good, I suggest to reconsider the hypotheses. Is it important to cut them into A and B parts? They focus on the same thing from the opposite angle.

Response: 

Thank you for your thoughtful comments.

In the research of economic empirical analysis, some researchers will directly put forward one hypothesis and some will put forward two opposite hypotheses. It is depend on the specific content of the research and literature review. Our research is similar to the latter. Hypothesis A and hypothesis B represent two aspects : null hypothesis and alternative hypothesis.

As we have discussed in literature review part, customer concentration has not only positive but also negative effects on enterprises (Certification effect and Concentrated credit risk). In that case it may have a "double-edged sword" effect on firm risk-taking.Therefore, it is more appropriate to write two Hypothesis A and hypothesis B. In addition, our finding "a nonlinear relationship between customer concentration and risk-taking, corporate policies and firm performance" also shows that these two effects exist in China.

4.Materials and methods: plain text is too much. Authors should try to visualize the content at least by using listed form.

5.Results: The content of S1 table should be inserted into the text. This table defines the terms which are used in the results chapter. Without that information the content is not clear.

Response: 

Thank you for your comments. We rewrote Section 3 and inserted S1 table into the text. We stated that “definitions of the variables are provided in the Table 1.” In that case, the content of main Materials and methods is much more clear. In addition, we deleted duplicate similar statements to show different measures as we mentioned briefly. Please see revised Section 3, on page 11-12.

6.The legend for the abbreviations shall be given below the tables. The format of the tables is not clear enough. The conclusions derived from the analyses should be highlighted at the end of the subchapters, at least in a new paragraph.

7.The asterisks ***, **, * are very confusing in the tables. Other indication should be used.

Response: 

We are very thankful for this comment. We are sorry that the format of the tables is not clear enough. The legend for the abbreviations have be given below the tables. The description of each table has been modified in revised paper. 

Another indication have been used to shoe the asterisks ***, **, * clearly. In the revised paper, we stated that “The standard errors are given in parentheses. Significance levels: ***p < 0.01,** p < 0.05, and * p < 0.1.”

Moreover, we are sorry that we do not highlight our finding. In the revised manuscript, we reorganized added more discussions of the unique results at the end of the subchapters (several new paragraphs) . Please see revised Section 4 and 5, on page 21 and 28.

8.In the present form it is hard to find what was the aim for conducting the performed analyses. Again, I recommend to add a visualized conceptual model which may help to follow the process of the research, how the analyses lead us to the new results.

Response: 

we added a figure to clearly clarify the economic logic framework of our research in the revised manuscript, Please see revised Section 1, on page 2-5,9.

 The economic channels or research logic frame is as follows:

Fig. 1. Logical diagram of customer concentration effect on firm risk-taking.

9.Conclusions: must be broadened. It says: "We empirically document that an inverse U-shaped curve exists between customer concentration and corporate risk-taking." Yes, but it is not highlighted in the results. Please show it.

Response: 

Thank you for comments. We made our best efforts to improve the writing. 

We rewrote the discussion of the main findings in all sections in the revised manuscript and added several new paragraphs. For example, we gave detailed discussion of in Section 4on page 24 and Section 5 on page 31 to highlight results. 

We also added more contents of conclusion in Section 6 and added further analyses in Section 6 For example, we argue that “The role of customer concentration could be treated as "double-edged sword" for the hosting firms”

Please see revised Section 6, on page 28 to 30. 

 It is seen that authors presented a complex research, but the way of presentation must be improved for acceptance.

Once again, thanks so much for your insightful comments and suggestions. They have helped us improve this paper significantly.

Response to Reviewer 2

Title:Does the risk of major customer need to be balanced? The role of customer concentration in corporate governance

(PONE-D-21-18289)

We appreciate the comments and suggestions in your referee’s report, which have helped us to better position the paper. We have carefully revised the paper to incorporate your suggestions. We believe that the incorporation of your suggestions has made significant improvement to this paper. The revised part of the paper is in blue. Below please find our point-by-point responses, with the referees’ original comments shown in italics. And the revised manuscript are attached.

Reviewer #2: The paper investigates the role of customer concentration on corporate governance and how the risks of main customers can influence the company.

The study is well written, its structure is perfect (except the keywords which are missing).

The abstract is well written, giving a brief highlight of the content in an excellent style.

The introduction explains the importance of the topic and introduces the importance and context, supported by some literature sources.

The literature review is well based, critical, and analytical enough; the hypotheses are developed in this framework too.

The methodology is clearly written, logical. Moreover, the authors selected the appropriate methodological toolset and applied them very well.

The methodology well supports the results; the conclusions are based on the results.

The language of the paper is scientific enough and well understandable.

.Response: 

Thank you for comments. 

In the submission system, the key words we write are as follows:

Keywords: corporate governance; customer concentration; risk-taking; monitoring and certification；firm performance

We also made our best efforts to improve the quality of the paper. 

First,we reorganized Introduction with 4 pages by abbreviating research contribution and deleted duplicate similar statements. 

Second, we added a new paragraph to explain the chapter arrangement and structure and a figure to clearly clarify the economic logic framework of our research in the revised manuscript.

Third, we highlighted our finding.In the revised manuscript, we added more discussions of the unique results at the end of the subchapters (several new paragraphs).We also added more contents of conclusion in Section 6. 

Fourth, we use a figure to show the economic channels or research logic frame in Section 3. 

Fig. 1. Logical diagram of customer concentration effect on firm risk-taking.

Once again, thanks so much for your insightful comments and suggestions. They have helped us improve this paper significantly.

---

## [Decision Letter · Decision Letter 1]

25 Oct 2021

Does the risk of major customer need to be balanced? The role of customer concentration in corporate governance

PONE-D-21-18289R1

Dear Dr. Ma,

We’re pleased to inform you that your manuscript has been judged scientifically suitable for publication and will be formally accepted for publication once it meets all outstanding technical requirements.

Kind regards,

László Vasa, PhD

Academic Editor

PLOS ONE

Additional Editor Comments (optional):

Reviewers' comments:

Reviewer's Responses to Questions

**Comments to the Author**

1. If the authors have adequately addressed your comments raised in a previous round of review and you feel that this manuscript is now acceptable for publication, you may indicate that here to bypass the “Comments to the Author” section, enter your conflict of interest statement in the “Confidential to Editor” section, and submit your "Accept" recommendation.

Reviewer #1: All comments have been addressed

Reviewer #2: All comments have been addressed

2. Is the manuscript technically sound, and do the data support the conclusions?

Reviewer #1: (No Response)

Reviewer #2: Yes

3. Has the statistical analysis been performed appropriately and rigorously? 

Reviewer #1: (No Response)

Reviewer #2: Yes

4. Have the authors made all data underlying the findings in their manuscript fully available?

Reviewer #1: (No Response)

Reviewer #2: Yes

5. Is the manuscript presented in an intelligible fashion and written in standard English?

Reviewer #1: (No Response)

Reviewer #2: Yes

6. Review Comments to the Author

Reviewer #1: (No Response)

Reviewer #2: The author made significant changes based on my previous recommendations and improved the paper well. Based on its current status and quality, I find it eligible for publishing in Plos One without changes.

7. PLOS authors have the option to publish the peer review history of their article (what does this mean?). If published, this will include your full peer review and any attached files.

Reviewer #1: No

Reviewer #2: No

---

## [Editor Report · Acceptance letter]

29 Oct 2021

PONE-D-21-18289R1 

Does the risk of major customer need to be balanced? The role of customer concentration in corporate governance 

Dear Dr. Ma:

I'm pleased to inform you that your manuscript has been deemed suitable for publication in PLOS ONE. Congratulations! Your manuscript is now with our production department. 

Kind regards, 

on behalf of

Prof. Dr. László Vasa 

Academic Editor

PLOS ONE